# New Developments in Material Preparation Using a Combination of Ionic Liquids and Microwave Irradiation

**DOI:** 10.3390/nano9040647

**Published:** 2019-04-22

**Authors:** Yannan Wang, Qidong Hou, Meiting Ju, Weizun Li

**Affiliations:** College of Environmental Science and Engineering, Nankai University, Tianjin 300350, China; wangyannan@nankai.edu.cn (Y.W.); houqidong@nankai.edu.cn (Q.H.); jumeit@nankai.edu.cn (M.J.)

**Keywords:** ionic liquids, microwave, synthetic methods, nanomaterials, polymers

## Abstract

During recent years, synthetic methods combining microwaves and ionic liquids became accepted as a promising methodology for various materials preparations because of their high efficiency and low energy consumption. Ionic liquids with high polarity are heated rapidly, volumetrically and simultaneously under microwave irradiation. Hence, combination of microwave irradiation as a heating source with ionic liquids with various roles (e.g., solvent, additive, template or reactant) opened a completely new technique in the last twenty years for nanomaterials and polymers preparation for applications in various materials science fields including polymer science. This review summarizes recent developments of some common materials syntheses using microwave-assisted ionic liquid method with a focus on inorganic nanomaterials, polymers, carbon-derived composites and biomass-based composites. After that, the mechanisms involved in microwave-assisted ionic-liquid (MAIL) are discussed briefly. This review also highlights the role of ionic liquids in the reaction and crucial issues that should be addressed in future research involving this synthesis technique.

## 1. Introduction

Green chemistry has received increasing attention during the past few decades because of energy crises and environmental pollution. The green chemistry principles led to the development of cleaner and more benign chemical processes, especially in chemical syntheses. Both ionic liquids (ILs) and microwave irradiation are known as a promising technology tool capable of contributing to green chemistry development because of their fast and easy applications, high efficiency and relatively environmental friendliness.

ILs are molten salts in a liquid form at low temperatures (usually below 100 °C) and are composed of large organic cations and small inorganic or organic anions. Comparing with conventional organic solvents, ILs have many unique properties, such as negligible vapor pressure, high temperature stability, non-volatility, chemical stability, large electrochemical stability window and ionic conductivity. Some ILs are also relatively environmentally friendly solvents compared with organic solvents because of their low vapor pressures especially when taking in account volatile organic compounds (VOCs) [1,2,3,4,5,6,7,8,9].

There are currently three different generations of ILs that have been identified. The first generation of ILs were mainly composed of cations like dialkylimidazolium or alkylpyridinium derivatives and anions like chloroaluminate or other metal halides, which have been described as toxic, non-biodegradable and oxygen-sensitive [10]. After that, the water and oxygen sensitive anions were replaced by halides (Cl^−^, Br^−^, I^−^) or anions such as BF_4_^−^, PF_6_^−^ and C_6_H_5_COO^−^, which are stable in water and air. These ILs possess lower melting points, different solubilities in classic organic solvents, and different viscosities, with an important disadvantage of low toxicity. This generation of ILs attract wide attention and provide interesting and novel application in materials areas [10]. The third generation of ILs named deep eutectic solvents (DESs) based on more hydrophobic and stable anions such as [(CF_3_SO_2_)_2_N^−^], sugars, amino or organic acids, alkylsulfates or alkylphosphates, and cations such as choline [11]. DESs are formed by hydrogen bond acceptors and hydrogen bond donors, which can be associated with each other by means of hydrogen bond interactions. Due to their low cost and analogous physico-chemical properties, DESs have attracted considerable attentions in recent years [12,13,14]. Generally, ILs can be synthesized by chemical synthesis, electrochemical synthesis, microwave-assisted synthesis and ultrasound-accelerated processes [15]. Because of variety of cations and anions, approximately 10^18^ different combinations are possible for IL syntheses [3]. Therefore, ILs will always be referred to as “designed liquids” with properties able to be tuned by adjusting their cations and anions [16,17,18,19,20]. In other words, ILs can be tweaked to meet different requirements by changing types of anions and cations used. For instance, the combination of an imidazolium cation with Cl^−^ or BF_4_^−^ results in a hydrophilic IL, whereas its combination with PF_6_^−^ results in a hydrophobic IL [17]. Furthermore, ILs containing metal speciations demonstrate additional functionalities such as magnetic, optical or catalytic activities [21,22]. Thus, ILs demonstrate wide applicability to the synthesis of various materials.

On the other hand, consumption of energy for heating is also an integral part of any chemical process. Microwave irradiation is one of the highly desirable energy sources to overcome this problem. Microwave irradiation refers to high frequency electromagnetic waves with wavelengths ranging from 1 mm to 1 m and alternating current signals with frequencies in the 0.3–300 GHz range. Exposure to microwaves results in rapid volumetric heating without the heat conduction process, thus, uniform heating can be achieved in a short period of time [23,24,25,26,27,28]. In comparison with traditional heating methods, not only can microwave heating rapidly provide high temperature and pressure and significantly decrease reaction time in a closed system but also can replace many organic solvents with water or even enable the reactions to proceed under solvent-free conditions with maximum efficiency [29,30]. Microwave heating provides several additional reaction parameters including power of the microwaves, combination of various organic solvents or even aqueous systems [31,32], inert atmosphere, etc., all of which provide possibilities to obtain a variety of morphologies [33]. Taking advantages of all the above-mentioned benefits, microwaves are widely used not only for common reactions such as materials syntheses, polymerization reactions and biomass extraction but also for some unique process such as hierarchical self-assembly of nanomaterials [32,34], etc.

ILs consist of organic multi-ring and/or organic alkane chains and bulky inorganic anions with high polarizability in a microwave field, which enables them to be heated uniformly and quickly throughout the whole reaction volume. Therefore, ILs are known as excellent microwave absorbers [35]. The first usage of ILs in a microwave-assisted process was reported by Varma and Namboodiri in 2001 [36]. Since then, an ever-increasing number of studies on microwave-assisted ionic-liquid (MAIL) methods were reported. The combination of microwaves and ILs delivers advantages of both, which provides a great potential to meet an ever increasing demand for effective, non- or low-toxicity and economic chemical processes [37].

This paper reviews recent progress on applications of the microwave/ILs technique for rapid and eco-friendly syntheses of functional materials. Preparation of inorganic nanomaterials (such as metal nanoparticles, metal oxides and other complex metal structures using this method) is discussed first. Then, an overview of the polymerization reactions conducted using MAIL methods is provided, followed by a research overview on syntheses of carbon-derived composites. Finally, a review of preparation techniques and various applications of biomass-based nanocomposites and biomass extraction products concludes this paper. The mechanisms and prospects of this method are also discussed briefly.

## 2. Preparation of Inorganic Nanomaterials

Over the past few decades, inorganic nanomaterials science experienced exponential growth in various fields because of unique physical and chemical properties of nanomaterials [38]. Properties of inorganic materials often depend on their morphology and dimensionality, thus, a lot of strategies to control shape of such nanomaterials were developed and reported in the literature [39]. ILs are widely used for inorganic material syntheses as microwave absorbers, solvents, additives and even as templates to optimize synthesis parameters.

### 2.1. Metal Nanostructures

The very first report on the preparation of inorganic nano-metals was published by Zhu et al. in 2004 [40]. They synthesized single-crystalline tellurium nanorods and nanowires by MAIL method at 180 °C using a N-butylpyridinium tetrafluoroborate ([BuPy][BF_4_]) IL. The whole synthesis took only 10 min. In this work, they demonstrated that combination of an IL and microwave heating played an important role in the synthesis of these metal nanostructures. When conventional heating methods (e.g., oil bath) and other solvent were used, fewer or no nanostructures formed. Other studies also reported successful syntheses of nanostructures of Au, Pt and Rh using the MAIL technique [41,42,43,44,45,46,47,48,49,50,51]. Many researchers emphasize that ILs act as steric stabilizers, solvents and/or microwave absorbers during synthesis of metal nanostructures [41].

Gold nanostructures are the most studied metal nanomaterials because of their wide use. Li et al. synthesized large-size single-crystal gold nanosheets by microwave heating of HAuCl_4_ in a 1-butyl-3-methylimidazolium tetrafluoroborate ([Bmim][BF_4_]) IL for 10 min without any additional template agents [42]. These Au nanosheets were over 30 μm in size. Most recently, Bhawawet et al. designed colloidal gold nanoparticles capped with oleylamine to use them for synthesis of gold nanoparticles by the MAIL method (Figure 1) [43]. They demonstrated shorter reaction times (from 30 s to 1 h) when microwaves were used compared to conventional methods. Their MAIL reactions involved toluene and 1-butyl-1-methylpyrrolidinium bis(trifluoromethylsulfonyl) imide ([C_4_mpy][Tf_2_N]). Usage of this IL as reaction solvent increased polydispersity of Au nanoparticles (NPs) and reduced reaction time. They also demonstrated that their ILs could be reused and could provide the same reaction effect as fresh ILs.

Li et al. synthesized Ag nanoparticles by a microwave-assisted hydrothermal technique using 1-dodecyl-3-methylimidazolium chloride ([C_12_mim]Cl) [44]. Their transmission electron microscopy (TEM) showed spherical Ag NPs ~12 nm in size on average. Morphology of Ag NPs was homogeneous. Additionally, particles were crystalline and showed no significant agglomerations. All these factors were attributed to IL participation during Ag NP synthesis. Coulomb coupling forces between Cl^−^ and [C_12_mim]^+^ ions resulted in their alignment, which, in turn, ensured an arrayed structure of the final reaction product. These Ag NPs improved cell activation properties.

The MAIL method was also used for synthesis of other noble metal nanoparticles. For example, Safavi synthesized spherical Pt and Pd nanostructures using various IL media [45]. Schütte et al. prepared Cu, Zn and Cu/Zn brass alloy nanoparticles from metal amidinate precursors using [Bmim][BF_4_] [46]. Another study also reported the important role of [Bmim][BF_4_] in the synthesis of transition-metal nanoparticles (M-NPs) (M = Ni, Cu). [47].

Carbon materials (e.g., graphene) are well-known substrates for metal NP deposition [48]. Synthesis of metal NPs on such materials can also be implemented using ILs and microwaves. Thus, Lee et al. synthesized Pt NPs on graphene/graphite oxide using 2-hydroxyethanaminium formate ([HOCH_2_CH_2_NH_3_][HCO_2_]) in combination with microwaves [49] and obtained structures with catalytic hydrogenation capability much higher than commercially-available catalysts. Esteban et al. synthesized stable hybrid iridium@graphene (Ir@TRGO) nanomaterials in a [Bmim][BF_4_] IL also using microwave irradiation [50]. Ir@TRGO nanomaterials with average particle diameters <5 nm were used as catalysts of benzene and cyclohexene hydrogenation.

### 2.2. Metal Oxide Nanostructures

#### 2.2.1. Zinc Oxide (ZnO)

ZnO is a wide bandgap semiconductor with a variety of practical applications. Wang et al. was the first research group to report ZnO synthesis using the MAIL method [51]. They obtained flower- and needle-like ZnO using [Bmim][BF_4_] as an additive upon exposure of the whole synthesis system to microwaves for only 5–10 min. They observed that the morphology of the resulting product was dependent on the temperature of the microwaves and IL amount used during the synthesis. Morphology of ZnO changed between flower-like and needle-like by changing the reaction conditions (Figure 2). Flower- and shuttle-like ZnO nanomaterials were later synthesized by the MAIL method using 1-butyl-3-methylimidazolium chloride ([Bmim]Cl) and [Bmim][BF_4_] ILs [52]. It was concluded that besides shorter reaction time, movement and polarization of ions under microwave irradiation resulted in the formation of transient and anisotropic micro-domains in the system, which led to the formation of ZnO with different morphologies. Thus, both the IL and microwave irradiation played an important role in the generation of ZnO with different morphologies.

Because of their favorable bandgap values, ZnO-based materials are usually used as photoanodes and photocatalysts. Rabieh et al. synthesized photocatalytic ZnO NPs using a [Bmim]Cl IL and 400 W microwave irradiation for only 1 min [53]. In this case, the IL acted as a surface agent and as a microwave absorbent. The resulting spherical ZnO NPs showed excellent photocatalytic performance toward photodegradation of the malachite green. Esmaili et al. also prepared nanocrystalline ZnO in a 1-ethyl-3-methylimidazolium ethylsulfate ([Emim][EtSO_4_]) IL for methylene blue (MB) photodegradation [54]. They proposed that using the IL as a reaction medium could decrease ZnO crystalline size and improve its nucleation rate, which eventually would lead to increased photocatalytic activity. One example using ZnO as a photoanode for solar cells is sensitized ZnO particles prepared with different amounts of a 1,3-dimethylimidazolium iodide ([Dmim]I) IL (da Trindade et al.) [55]. In their system the IL was a dry solid. It did not change ZnO phase formation and/or morphology, but decreased ZnO particle size and converted shallow defects into deep ones, which led to a significant increase of photocurrent density.

#### 2.2.2. Titanium Dioxide (TiO_2_)

TiO_2_ is most frequently used as a photocatalyst because of its low cost, non-toxicity, high activity and stability. TiO_2_-based nanomaterials with different morphologies were fabricated to enhance photocatalytic activity of TiO_2_ under visible light. Some studies reported that ILs could facilitate formation of hollow TiO_2_ [56], self-assembled TiO_2_ nanosheets [57], mesoporous TiO_2_ [58], etc.

Recent research reported synthesis of TiO_2_ with higher photocatalytic activity using microwave irradiation combined with ILs. Zhang et al. first proposed synthesis of micro-sheet anatase with reactive (001) facets using microwave and a [Bmim][BF_4_] IL [59], as shown in Figure 3. The IL in their study provided a fluorine-rich environment and helped to expose (001) facets. Usage of microwaves decreased reaction time and increased crystallinity of TiO_2_. As a microwave absorbent, the IL also enhanced microwave efficiency. Chen et al. also mentioned that [BF_4_]^−^ anion in the IL was a critical factor for the synthesis of Ti^3+^ self-doped TiO_2_ hollow nanocrystals under microwave irradiation [60]. They used [Bmim][BF_4_] to take advantage of its microwave absorbing and morphology-controlling properties. As a result, they obtained TiO_2_ mesoporous nanocrystals. It was suggested that the IL also facilitated decreasing recombination of photoelectrons and holes, which normally favors high photocatalytic activity [61].

Xiao et al. synthesized in situ carbon nanotubes (CNTs) threaded with TiO_2_ single crystals with exposed (001) active facets (CNTs–TiO_2_) in a 1-methyl imidazolium tetrafluoroborate ([Hmim]BF_4_) IL using microwave irradiation [62]. The system was heated at 13 °C min/L to 150 °C and kept at that temperature for 30 min. Field emission scanning electron microscope (FESEM) images of the resulting CNTs–TiO_2_ structures with different initial concentrations of TiCl_3_ are shown in Figure 4. To prepare CNTs–TiO_2_-4 (see Figure 4), which demonstrated the highest photocatalytic activity for NO removal, 0.5 M of TiCl_3_ was used as a precursor. They suggested that presence of [BF_4_]^−^ groups was critical for the formation of a photocatalyst with such high activity. These groups (1) assisted rutile→anatase phase transition, (2) coordinated with Ti^3+^ to oxidize TiO_2_ crystals and (3) helped formation of decahedrons with exposed (001) facets. Microwaves helped to increase the surface of the “super-hot” dots to accelerate formation of TiO_2_ crystals.

#### 2.2.3. Other Metal Oxide Nanomaterials

CuO is a narrow bandgap transition metal oxide widely used for photoconductive and photochemical applications [63]. CuO synthesis by MAIL methods can change the structure of nano-CuO, improve its performance or increase the reaction efficiency. Different morphologies of CuO were prepared by MAIL methods: Nanosheets and nano-whiskers [64], flower- and leaf-like nanomaterials [65], feather- and flower-like crystals [66], CuO NPs [67] and CuO/Cu_2_O composites [68].

Flower-like Cu_2_O with high photochemical activity also can be prepared by the MAIL method [69] often using 1-octyl-3-methylimidazolium trifluoroacetate ([Omim][TA]) and [Bmim][BF_4_] ILs. ILs again played an important role in controlling Cu_2_O-based nanomaterials morphologies, which are needed to enhance Cu_2_O performance in different applications. Xia et al. achieved different morphologies of CuO crystals (flower-like to leaf-like) by controlling the concentration of an [Omim][TA] IL with high ionic conductivity and polarizability (see Figure 5) [65]. The first step of their morphology-controlling mechanism involved microwave energy absorption by [Omim]^+^, which led to high heating rate and shorter reaction time. The second step was weak long-range ordering of [Omim][TA] at higher concentrations, which prevented particles accumulating and forming ordered and/or organized structures [70]. Hence, different concentrations of IL resulted in the formation of CuO structures with different morphologies.

Iron oxides are functional materials of great industrial and scientific importance because of their chemical stability and non- or low-toxicity. Ahmed et al. fabricated α-Fe_2_O_3_ quantum dots using a [Bmim][BF_4_] IL and a microwave-assisted solution synthesis method [71]. These quantum dots were ~10–13 nm in diameter with the outstanding photocatalytic properties. Cao and Zhu reported a microwave- and IL-assisted hydrothermal method for the synthesis of α-Fe_2_O_3_ and Fe_3_O_4_ NPs with good photocatalytic activity [72]. They demonstrated that it was indeed the [Bmim][BF_4_] IL that influenced phase composition and morphology of the final product. Yin et al. showed that temperature and time of microwave exposure also influenced morphology of Fe_3_O_4_ during MAIL synthesis with a [Bmim][BF_4_] IL [73]. Fe_3_O_4_ nanowires were obtained at 70 °C while nanorods were obtained at 90 °C.

The MAIL method was also successfully used to synthesize other metal oxide nanostructures. Jadhav et al. obtained controlled MgO nanostructures with various green IL solvents using microwave irradiation [74]. Their ILs were N-methyl imidazolium and 3-methyl pyridinium cations with various halide anions. They used ILs not only as solvents but also as structural directing agents. MgO nanostructures with various morphologies were obtained using different ILs and MW conditions (Figure 6). SnO, a very popular anode material, can also be synthesized by the MAIL method. Qin et al. synthesized SnO at low temperature with excellent cycling performance using 1-ethyl-3-methlyl-imidazolium acetate ([Emim][Ac]) and microwave irradiation [75]. ILs played a role of a microwave absorbent to reduce reaction temperature and time.

#### 2.3.4. Other Complex Metal Structures

Using a combination of microwave irradiation and ILs, other complex metal structures with enhanced performance were synthesized. Cybinska et al. prepared Ln^3+^-doped (Ln = Sm, Eu, Tb, Dy) BiPO_4_ NPs with particle sizes <10 nm from the respective metal acetates and choline or butylammonium dihydrogen-phosphate in a microwave at 120 °C for 10 min [76]. In their work, ILs acted as a solvent, microwave susceptor and reaction stabilizer; they also affected particle morphology and size. A big advantage of MAIL methods is that no post processing (including heat treatment) of the final product is needed, which helps to avoid particles aggregation.

Siramdas et al. successfully synthesized luminescent InP nanocrystals with different sizes using a 1-butyl-4-methylpyridinium tetrafluoroborate ([BmPy][BF_4_]) IL [77]. The resulting products exhibited color-tunable luminescence in the visible region with 20%–30% quantum yields. They compared products obtained without an IL in the reaction system (see Figure 7) and concluded that IL presence was a key factor responsible for efficient nanocrystals luminescence. This simple procedure provides a pathway to obtain luminescent nanocrystals using a simple preparation method with just a single precursor.

Olchowka et al. fabricated fluoridosilicates A_2_SiF_6_ (A = Li, Na, K, Rb, Cs) by the MAIL method using 1- butyl-3-methylimidazolium hexafluorophosphate ([Bmim][PF_6_]). These materials are promising host lattices for LEDs [78]. During their synthesis, an IL was used as a low-temperature solvent and as a fluorine source, which helped to eliminate usage of hazardous HF. They suggested expanding usage and variety of different ILs as well as reaction time to explore this green method to obtain a variety of fluoride-containing LED host materials with different particle sizes and shapes as well as functionalities.

Various ILs were used by Alammar et al. to synthesize nano-SrSnO_3_ photocatalysts using microwaves [79]. They used ILs with different cations (e.g., 1-butyl-3-methylimidazolium ([C_4_mim]^+^), 6-bis(3-methylimidazolium-1-yl)hexane ([C_6_(mim)_2_]^2+^), butylpyridinium ([C_4_Py]^+^), tetradecyltrihexylphosphonium ([P_66614_]^+^)) and bis(trifluoromethanesulfonyl)amide ([Tf_2_N]^−^) as an anion. They concluded that usage of different ILs could lead to products with different morphologies and sizes. They also emphasized, that unlike the IL cation, the choice of anion had little effect on morphologies of the formed products. 

Besides the mentioned references above, Table 1 lists an updated review of other metal structures prepared by microwave-assisted ionic-liquid (MAIL) methods reported in the literature.

## 3. Polymers Preparation

In order to reduce consumption of volatile organic compounds (VOCs), the polymer industry is in search for alternatives, some of which are water or supercritical CO_2_ [88]. However, some polymer-related processes cannot be exposed to moisture, and practices involving supercritical CO_2_ require high pressure [89]. Another alternative is the implementation of ILs as polymer reaction media. Polymerization reactions conducted in ILs exhibited higher polymerization rate and higher molar weight of the resulting polymers. The application of microwave irradiation in the procedures involving polymers also attracts attention because of its higher efficacy compared to conventional heating methods. Thus, the combination of ILs and MW could promote development of greener and more efficient polymer reactions with significant energy savings as well as with reduced or completely eliminated VOCs [90].

### 3.1. Ring Opening Polymerization

Cationic ring opening polymerization (CROP) is an easy route to synthesize poly(2-oxazoline)s (POxs) [91,92,93]. POxs are functionalized polymers with various applications, however, they often have low polymerization rates and high VOC emissions during their production which limits their vast development [94,95,96]. The first study showing the application of ILs as reaction media for CROP under MW was reported by Guerrero-Sanchez et al. [97]. Even though they performed CROP of 2-ethyl-2-oxazoline in different ILs, their main focus was CROP in a [Bmim][PF_6_] IL because poly(2-ethyl-2-oxazoline) is more soluble in water than an IL which can improve polymer isolation and IL recycling. The polymerization process was finished within 1 min under MW, and faster polymerization rate and lower polydispersity was observed during this reaction in comparison with reactions using other common organic solvents. Additionally, the [Bmim][PF_6_] IL was efficiently recovered by water extraction and reused with similar results. They also successfully synthesized 2-phenyl-2-oxazoline and 2-(m-difluorophenyl)-2-oxazoline in water-soluble 1-butyl-3-methylimidazolium trifluoromethanesulfonate ([Bmim][TfO]) and [Bmim][BF_4_] ILs in just 30 min by microwave-assisted reaction [98]. Their results showed that polymers obtained in ILs demonstrate broad molecular weight distributions and higher polydispersity.

Poly(ε-caprolactone) (PCL) is one of the most common industrial biodegradable materials for bio- and medical packaging applications. Ring opening polymerization (ROP) of ε-caprolactone(ε-CL) to obtain PCL requires over 24 h when conventional heating is used [99]. Liao et al. obtained PCL in just 30 min by performing ROP with ZnO as a catalyst and under the presence of [Bmim][BF_4_]. They observed that at the same microwave power, temperature of the reaction mixtures can be increased by just varying concentration of [Bmim][BF_4_]. Additionally, ROP of ε-CL was also successfully performed without a catalyst but with [Bmim][BF_4_] at higher temperature. The same team studied microwave-assisted ROP of another biodegradable material—poly(trimethylene carbonate) (PTMC) [100]. When 5 wt.% of [Bmim][BF_4_] was added, PTMC with molar weight equal to ~36,400 g/mol was obtained at 5 W microwave power in 60 min. Molar weight and reaction rate were both increased when the combination of ILs and MW were used.

### 3.2. Polycondensation

Polycondensation reactions using the MAIL technique were mainly reported by Mallakpour’s team. They prepared chiral poly(amide–imide) nanostructures, active poly(ester–imide) and other polyamides [101,102] using tetrabutylammonium bromide (TBAB), 1,3-diisopropylimidazolium bromide([1,3-(isopr)_2_im]Br) ILs, etc [103]. In a typical reaction, 5-(2-Phthalimidiylpropanoyl-amino) isophthalic acid, diamines [1,3-(isopr)_2_im]Br and 1,5-Naphthalenediamine or other aromatic diamines were placed in a porcelain dish and then triphenyl phosphite (TPP) was added [104]. The mixtures were heated under microwave irradiation for a total of 90 s. After filtering and drying, the yield was 93%. They also compared their MAIL methods to conventional heating ones. Products obtained under microwave irradiation had remarkable solubility, optically activity and good thermal stability. At the same time, reaction time was reduced remarkably, and ILs used were harmless green solvents. Thus, the techniques developed by Mallakpour’s team are environmentally friendly polycondensation reactions.

### 3.3. Free Radical Polymerization

During heterogeneous free-radical polymerization (FRP) reactions, ILs can also act as reaction media, as surfactants and as microwave absorbents. Guerrero-Sanchez et al. first investigated FRP of methyl methacrylate in [Bmim][TfO] and [Bmim][BF_4_] ILs under microwave irradiation [98]. ILs in this reaction provided a more efficient absorption of microwave irradiation and increased the heating rates. At the same time, because of the dispersed phase, type of IL and its concentration also influenced yield and molecular weight of the resulting polymer. After the reaction, ILs were recovered and reused, which confirmed that MAIL-based FRP reactions are not only efficient but also environmentally friendly.

Glück’s team studied systematic variation of MAIL FRP reaction conditions to compare them with conventional heating methods and solvents [105,106]. They found that when conventional solvents (such as N,N-Dimethylformamide(DMF) or methanol) were used, conversion rates of the FRP reaction conducted using microwaves and using conventional heating were the same. However, when ILs were used as solvents, FRP conversion rate upon microwave heating was higher relative to conventional heating whether with or without ILs. Polymerization rate r_p_ was higher in ILs than in conventional solvents upon microwave irradiation but only for some reactions. In other reactions, r_p_ was nearly the same in ILs and in DMF/methanol. These differences might be because of complexity of polar interactions between ILs and the monomer and/or polymer radicals.

### 3.4. Copolymers

#### 3.4.1. Cellulose Graft Copolymers

Intelligent polymers with unique properties in response to external triggers are attractive for biotechnology. Grafting such intelligent polymers on cellulose was extensively studied. Using dissolving capacity of cellulose in [Bmim]Cl, Wei et al. grafted 2-(Dimethylamino) ethylmethacrylate (DMAEMA) onto bagasse cellulose by microwave heating [107]. This cellulose-g-DEAEMA copolymer was sensitive to pH and salinity in an acidic environment or a low salt concentration. A homogeneous cellulose–acrylic acid graft copolymer as an effective metal ion adsorbent was prepared by Lin et al. [108] using [Bmim]Cl as a powerful direct solvent. Usage of microwave irradiation reduced the time of this reaction to only 3 min. After optimization of reaction parameters (such as monomer concentration, exposure temperature and overall reaction time), a graft copolymer with excellent adsorption ability towards Cu^2+^ (to remove it from industrial waste and contaminated waters) was obtained.

#### 3.4.2. Copolyurethanes

Segmented polyurethanes (PUs), used in a variety of practical applications, in which structure variability is needed, are copolymers with hard and soft segments. Rafiemanzelat et al. synthesized novel multi-block polyurethane copolymers, containing ecofriendly and biodegradable segments, by two-step polymerization in TBAB under microwave irradiation [109]. First, 4,4′-methylene-bis-(4-phenylisocyanate) was reacted with l-leucine anhydride cyclodipeptide to produce isocyanate-terminated oligo(imide–urea) as a hard segment. Then, polyethylene glycol with a molecular weight of ~1000 was used to extend the chain of the pre-polymer for further synthesis of PU. Comparing to conventional heating methods, PUs prepared by the MAIL technique showed more phase separation and crystallinity. Furthermore, reusability of ILs was also confirmed in their work.

### 3.5. Poly(ionic Liquid)s

Poly(ionic liquid)s (PILs) can be synthesized by polymerization of IL monomers. PILs have a lot of unique properties: They inherit some properties of ILs (such as low vapor pressure and high thermostability) and possess well-defined solid morphologies and mechanical characteristics [110] of the original polymers. Although conductivity of PILs decreases after polymerization, PILs are still suitable for electro-responsive fluid applications because of locally-moving ions and the high density of cation/anion parts [111,112,113]. Dong et al. prepared uniform P[MTMA][TFSI] (MTMA = [2-(methacryloyloxy)ethyl]trimethylammonium^+^, TFSI = bis(tri-uoromethane sulfonyl)imide^−^) particles with low density under microwave irradiation for 7 h at 70 °C (see Figure 8) [112]. Their PILs showed high electro-responsive fluid activity without the need for an activator and low leaking current density because of their hydrophobicity. Analysis of dielectric spectra showed that high density of cation/anion parts in PILs played an important role in strong dielectric polarizability and adequate polarization response of PILs. Overall, PIL particles have a lot of potential applications in electro-responsive systems. Based on previous research [105], Dong et al. further demonstrated electrorheological applications of PIL particles as well as studying how sizes of cation/anion parts influence electrorheological performance [113]. Monodisperse PIL particles with a polystyrene backbone and quaternary ammonium/fluorinated imides as ion parts were synthesized under microwave irradiation. Electrorheological effect was stronger when cation/anion parts were smaller, probably because of the higher density of ions and faster ion mobility rates.

Based on the hydrophilicity and good extraction efficiency of some PILs, Yang’s team designed. PIL-modified magnetic nanomaterials under microwave assistance for pesticide and phthalate esters extraction [114,115]. PIL (1-vinyl-3-butylimidazolium bromide)-functionalized modified polystyrene magnetic nanospheres (PILs-PMNP) were synthesized by a one-pot copolymerization reaction under microwave irradiation for 4 h. PILs-PMNPs were designed for magnetic solid-phase extraction (SPE) to extract and concentrate phthalate esters from beverages. These PILs-PMNPs demonstrated outstanding adsorption ability toward traces of PAE analytes compared with polystyrene-modified magnetic nanospheres. Yang et al. also designed PIL-modified magnetic nanoparticles based on a dispersion SPE method, which was combined with HPLC–UV to detect trace levels of pesticides in fruits and vegetables [115]. PIL materials could also be reused multiple times.

## 4. Preparation of Carbon-Based Composites

Carbon-based composites are widely used as catalysts, electrode and luminescent materials as well as super capacitors [21]. Carbon-based composites are typically synthesized by pyrolysis of raw biomaterials or polymers at high temperatures [116,117]. IL-based approaches to prepare carbon-based composites also attracted attention [118] mostly because of the unique properties of ILs [119,120]. For instance, ILs could be used as solvents for the synthesis of carbon materials from biomass [121], or IL cations or anions could act as porogens to create porous carbons [122], or even as carbon or nitrogen sources to produce carbon-based composites [123]. All these and similar methods when combined with microwave irradiation offer even more effective and economic methods to develop and design various carbon-based composites.

### 4.1. Carbon Dots/Nanodots

Carbon dots (CDs) have excellent luminescent properties, good solubility and biocompatibility because of their quantum properties [124]. Conventional CD syntheses usually occur at high temperatures. Usage of ILs could offer a larger temperature range for CD synthesis because of the unique physical and chemical properties of ILs [125]. ILs can also be used as solvents, carbon sources, modifiers for CD syntheses, etc. Microwave irradiation, which could be combined with ILs, can also be implemented to synthesize CDs especially with biomaterials as initial compounds [123]. Jeong et al. reported a novel approach to produce CDs from biomaterials [126] using just microwave irradiation: They fabricated CDs based on regenerated cellulose (RC–CDs) from recycled paper. They combined microwave irradiation with a 1-allyl-3-methylimidazolium chloride ([Amim]Cl) IL to extract cellulose from the recycled paper, which was then dissolved in the IL. Cellulose was then regenerated by adding absolute ethanol, after which the IL was recovered. RC–CDs exhibited low cytotoxicity and excellent fluorescence properties for potential bio-applications.

Rice straw, a common bio-resource, was also used as a raw material to fabricate photo-luminescing CDs by the MAIL method with an [Amim]Cl IL [127]. The IL helped to extract and dissolve cellulose from the rice straw and provided nitrogen as a heteroatom to obtain N-doped CDs. The resulting CDs were spherical and had a high quantum yield. Moreover, they were also used as efficient and very responsive Fe^3+^ sensors with a very low detection limit.

Very recently, Pham-Truong et al. used glutamine and glucose as carbon and nitrogen sources, respectively, to synthesize nano-CDs by the MAIL method using a 1-ethyl-3-methylimidazolium ethylsulfate ([Emim][EtSO_4_]) IL [125]. The resulting products demonstrated presence of pyridinic and graphitic nitrogen as well as imidazolium and the IL anion on their surfaces. The source of graphitic nitrogen was from the imidazolium ring in the IL. Oxygen reduction measurements showed that both CDs have promising electrocatalytic activity. However, CDs generated under the presence of glucose and the IL exhibited the highest efficiency (up to 90%) towards H_2_O_2_ production.

Liu et al. also used ILs as a carbon source to prepare fluorescence CDs by one-step microwave-assisted pyrolysis of a mixture containing a 1-vinyl-3-aminopropyl imidazolium ([PAVIm]Br) IL and ethylenediamine [128]. During their synthesis, the IL acted as a carbon source and ethylenediamine served as a nitrogen source for doping and as a surface passivating agent. The resulting CDs exhibited outstanding photoluminescence properties useful for detection of Cr(VI) as well as for temperature and pH values. Safavi et al. suggested adopting the MAIL method for a simple and efficient preparation of blue-luminescing CDs with narrow particle size distribution [129]. They used a N-octylpyridinum hexafluorophosphate IL as a microwave absorber and as a carbon source. The same CDs were used for effective catalytic degradation of azo dyes.

### 4.2. Carbon Nanoparticles

Carbon nanoparticles (CNPs) are other fluorescent materials used in many applications because of their outstanding properties such as fluorescence, low cytotoxicity and excellent biocompatibility. Xiao et al. group was the first to report microwave-assisted one-step synthesis of CNPs derived from ILs. Their CNPs were monodisperse with strong fluorescence and good water solubility [130]. They employed a [Bmim][BF_4_] IL as a carbon and nitrogen source as well as different microwave irradiation times to design various CNPs. The whole process was eco-friendly and economical without any complex procedures or toxic reactants. These as-prepared CNPs were incorporate into a novel fluorescence probe for quercetin detection. Wei et al. fabricated a CNP/ionic liquid (CNPIL) hybrid by exposing 1-butyl-3-methylimidazolium glutamine salt (as the IL) and glucose to microwave irradiation [131]. The resulting CNPIL showed excellent conductivity and dispersibility in organic and/or aqueous solvents. These CNPIL also demonstrated outstanding photoluminescence properties adjustable by specific synthesis conditions.

### 4.3. Other Carbon Materials

Meng et al. used IL 1-butyl-3-methylimidazolium dicyanamide ([Bmim][N(CN)_2_]) as a carbon source to prepare a N-doped carbon membrane under microwave irradiation. This membrane had high discharge capacity and excellent cyclic performance especially relative to membranes synthesized from glucose as a carbon source [132]. Because of the absorption properties of the IL, a uniform and network-like N-doped carbon membrane was coated on the surface of LiFePO_4_ to improve its electronic conductivity. Vadahanambi et al. reported a one-pot synthesis of holey-carbon nano-sheets (hCNS) by microwave carbonization of 1-ethyl-3-methylimidazolium tetrafluoroborate ([Emim][BF_4_]) in a soft-template [133]. The resulting hCNS were homogeneously distributed spheres 10–20 nm in diameter, which could be used as potential anode materials for sodium ion batteries.

MAIL technology can also be applied to modify carbon materials. Beggs et al. improved interfacial adhesion of carbon fibers by treating them in the (1-ethyl-3-methylimidazolium bis(trifluoromethylsulfonyl)imide) IL under lower-power microwave irradiation (20 W) [134]. Myriam et al. reported modification of CNTs by a newly-developed approach for controlled functionalization/retro-functionalization of single-walled CNTs using the MAIL method. This method can be used as a general technique to restore π-conjugated structure of carbon nanotubes and other carbon allotropes [135].

## 5. Preparation of Biomass-Based Materials

It is widely known that cellulose dissolves in ILs, and microwave heating can significantly accelerate the dissolution. Synthesis of biomass-based nanocomposites and products using ILs significantly broadens biomass application range. Additionally, using biomass as a raw material satisfies green chemistry principles of sustainable resources.

### 5.1. Cellulose-Based Composites

Recently, cellulose-based composites attracted a lot of attention as green materials. These composites enrich the family of green materials and expand usage of biomass for high value-added applications. Combination of cellulose with metals yields even more outstanding properties useful for biomedical applications [136]. During the synthesis of cellulose-based composites, ILs not only act as templates and solvents, they also have outstanding contribution to the dissolving capacity of cellulose. Based on these properties Jia’s group first developed MAIL technology to synthesize cellulose-based nanocomposites [137]. They prepared cellulose/calcium silicate nanocomposites using fresh and recycled ILs. They used [C_4_mim]Cl as a solvent to dissolve microcrystalline cellulose and microwaved the solution to 110 °C for 20 min. Ca(NO_3_)_2_·4H_2_O and Na_2_SiO_3_·9H_2_O were added as precursors. SEM of the nanocomposites synthesized by fresh, recycled and twice-recycled ILs are shown in Figure 9. Sizes and microstructures of the final products were different depending on which IL was used. Their team also synthesized cellulose/CuO and cellulose/CaCO_3_ using MAIL methods [138,139] with [Bmim][BF_4_] and [Bmim]Cl ILs, respectively. Both of these ILs are excellent for dissolution of cellulose and as microwave absorbents. Microwave-heating time and IL:cellulose ratio significantly affected the size and morphologies of the nanocomposite. Cellulose/CaCO_3_ nanocomposites synthesized using this method demonstrated good biocompatibility, which is useful for biomedical applications. Thus, this synthetic method opens a new way for cellulose applications.

Sharma et al. [140] prepared superabsorbent nanocomposite from sugarcane bagasse (SCB), chitin and clay by the MAIL method with a 2-hydroxy ethyl ammonium formate (HEAF) IL. SCB and chitin were first suspended in separate HEAF solutions and then allowed to precipitate. After mixing appropriate amounts of initiator, monomer and crosslinker, they obtained a nanocomposite with over 1000% swelling degree upon microwave irradiation for 24 h. They also found that these nanocomposites exhibited high microbial resistance and recyclability with a swelling degree even greater after each recycle.

Wood fibers can also be used as raw material to fabricate transparent film composites. Lu et al. obtained a highly transparent all-cellulose film with less time and better performance (than conventional techniques) by using MAIL technology [141]. They used an IL as a solvent to disrupt stiff structure of the wood fibers: the IL broke hydrogen bonds between or/and inner cellulose chains. Meanwhile, microwave irradiation increased the temperature of the system very quickly during the synthesis, which resulted in higher efficiency.

Cellulose-ZnO nanocomposite [142], cellulose-SnS_2_ composites [143], cellulose/CaF_2_ nanocomposites and cellulose/MgF_2_ nanocomposites [144] were also successfully synthesized using the MAIL method. Thus, choice of IL and microwave conditions affect morphologies and properties of the resulting nanocomposites. This method of preparation of novel materials opens a lot of possibilities for current and future cellulose-based nanocomposites.

### 5.2. Production of 5-Hydroxymethylfurfural

An IL was also used in 5-hydroxymethylfurfural (5-HMF) production from biomass with high efficacy. 5-hydroxymethylfurfural (5-HMF) is one of the top bio-based platform compounds, which acts as an important intermediate product to obtain fine chemicals, pharmaceuticals and biofuels [145]. 5-HMF is also a key building block and a starting point for synthesis of 2,5-dimethylfuran, 2,5-diformylfuran, 2,5-furandicarbaldehyde and 2,5-furandicarboxylic acids (see Figure 10) [146]. 5-HMF can be obtained from fructose, glucose or cellulose by hydrolysis, dissolution and acid-assisted catalysis. However, the yield of 5-HMF when produced from biomass or cellulose is unsatisfactory. Thus, ILs can be used in 5-HMF production from biomass because of its attractive cellulose dissolving ability. In fact, MAIL methods for 5-HMF production from cellulose were indeed reported [147].

Early reports on the application of MAIL method for 5-HMF production were generated by Prof. Li’s and Qi’s team. They both used CrCl_3_ as a catalyst, 400 W MW as a heating source and [C_4_mim]Cl and [Bmim]Cl as ILs, respectively [148,149]. Their groups achieved 61% and 71% yields for 5-HMF, respectively. They used ILs as solvents for the acid-catalyzed dehydration. At the same time, chloride anions in ILs contributed to the isomerization of glucose into fructose. [C_4_mim]Cl also acted as a scavenger for water: Water is often responsible for low yields as it decomposes freshly-formed HMF during its synthesis. Compared to conventional heating methods, MW reduced reaction time to 30 s, which was possible because of excellent dielectric properties of ILs. Prof. Qi et al. [149] tested recyclability of the system and demonstrated that ILs/CrCl_3_ could be recycled without compromising its performance.

Usage of ILs as solvents implies that high amounts IL are often needed during the synthesis. Qu et al. proposed to use ILs as catalysts (instead of as solvents) to convert microcrystalline cellulose to 5-HMF to make the process more economical and environmentally friendly [150]. The highest yield of 5-HMF was 28.63% when 1,1,3,3-tetramethylguanidine tetrafluoroborate ([TMG][BF_4_]) was applied as a catalyst under microwave irradiation at 132 °C for 48 min. Very recently, Zhang et al. developed a process to produce 5-HMF generation from cellulose at mild temperatures (~80 °C) using microwave irradiation for 3 h [151]. Their catalyst was a fluorine anion-containing IL functionalized with biochar sulfonic acids (BCSA-IL-F_1–3s_), which can be simply synthesized by ionic exchange of 1-trimethoxysilylpropyl-3-methylimidazolium chloride (IL-Cl) grafted on the BCSA with CF_3_SO_3_H (HF_1_), HBF_4_ (HF_2_), HPF_6_ (HF_3_), respectively. The highest yield of 5-HMF was 12.70%–27.94%.

### 5.3. Production of Furfural

Furfural is an important high value-added platform chemical derived from lignocellulosic biomass, such as corncobs, bagasse, wood chips and switchgrass. Furfural has great platform potential for the simultaneous production of fuels including methyl-tetrahydrofuran and liquid alkanes [152]. Serrano-Ruiz et al. proposed an efficient pathway for the production of furfural from C5 sugars and lignocellulosic waste catalyzed by Brönsted acidic ionic liquids (1-(4-Sulfonylbutyl) pyridinium methanesulfonate) under microwave irradiation (100 W, 150 °C). The yield of furfural was 85% and 45% for C5 sugars and lignocellulosic waste, respectively [153]. After that, Zhang et al. studied the conversion of xylan and lignocellulosic biomass (corncob, grass and pine wood) into furfural catalyzed by AlCl_3_ in [Bmim]Cl at 170 °C for 10 s microwave. The highest furfural production was 84.8% for xylan and 16%–33% for lignocellulosic biomass. Meanwhile, [Bmim]Cl and AlCl_3_ could be recycled for four runs with stable catalytic activity [154].

Zhang’s work focused on the conversion of lignocellulosic biomass into HMF and furfural at the same time in [C_4_mim]Cl in the presence of CrCl_3_ under microwave irradiation [155]. Corn stalk, rice straw and pine wood treated under typical reaction conditions produced HMF and furfural in yields of 45%–52% and 23%–31%, respectively, within 3 min. da Silva Lacerda et al. also produced HMF and furfural from lignocellulosic materials with a yield of 53.24% at optimized condition [156]. The analysis result showed that when an additional 15 min ultrasonic pretreatment is also conducted, microwave treatment combined with stirring could result in the best production yields. The reports described above should be valuable to facilitate cost-effective conversion of biomass into biofuels.

### 5.4. Production of Reducing Sugar

Bio-ethanol energy, as a renewable resource, gets increasing attention recent years. Reducing sugar hydrolyzed from cellulose and hemicelluloses can be relatively easy fermented to bio-ethanol. The application of ILs and microwaves in biomass hydrolysis cannot only obtain a high yield of reducing sugar, but also achieves high efficiency. Chen et al. proposed an effective and profitable process with [Amim]Cl as catalyst to produce reducing sugars from the hydrolysis of straw under microwave heating [157]. The highest yield of glucose and xylose can attain 29.1% and 24.3%, respectively. Hou et al. also achieved high sugar yield by optimized microwave-assisted [TBA][OH] pretreatment [158]. They suggested that the combined effect leads to the violent deconstruction of lignin and hemicelluloses, the crystalline region broken and an eroded, pored and irregular micro-morphology, which contribute to the high sugar yield.

## 6. Mechanisms Involved in MAIL Methods

There are several mechanisms involved in MAIL methods for materials preparation. First of all, ILs efficiently absorb microwave energy through an ionic conduction mechanism, and thus are employed as solvents and co-solvents, leading to a very high heating rate and a significantly shortened reaction time. Yin’s group also revealed that even a small amount of IL could act as microwave absorber to markedly shorten the reaction time [73]. The above-mentioned results all demonstrate that the reactions could be completed in a few minutes by means of MAIL methods.

Second, ILs are always used as solvents in biomass-based materials and carbon materials syntheses mainly due to the cellulose dissolving capacity of ILs. For instance, [Bmim]Cl is usually applied in biomass dissolution. The imidazolium cation with positive charge acting as electron acceptor and the chloride anion with negative charge acting as electron donor, interact with oxygen and hydrogen of -OH bonding of cellulose, respectively and promote the dissolution of cellulose. The hydroxyl groups on the cation side chain of the ionic liquid along with the anion also interact with the hydroxyl groups of cellulose, which further weakens hydrogen bonding among the cellulose fibers. Ionic liquids weaken the hydrogen bonding in cellulose which results in the formation of amorphous regions with a lower degree of order in the cellulose structure. Dissolution of the amorphous region takes place initially, followed by reaction of ionic liquid with the crystalline part of cellulose [159]. Based on this property, various materials were obtained.

Third, ILs are often used as additives in metal structures preparation due to their templating effect. There are many examples that MAIL method could resulted in different morphologies of the obtained nanostructured materials above. Ionic liquids play a critical role in the formation of various structures. Among the reports, imidazolium rings caused great attention in such reactions. The extended hydrogen-bonding and π-π stack interaction of the neighboring imidazolium rings enable ionic liquid molecular recognition and self-assembly. As a “supramolecular” solvent, the self-assembled ability of the IL significantly influences structural orientation in the reaction [160]. Pang et al. proposed the formation of different morphologies from another perspective. These ionic liquids have a cation head and carbon chain group, therefore they have the aggregation behavior similar to the surfactant. The aggregates in the solution might also have a function like the soft template for the growth of the inorganic crystals [161]. Microwave irradiation in some reports accelerates the migration rate of atoms which contribute to different structures compared with conventional heating method.

Last, parts of ILs also provide carbon, nitrogen and fluorine resources to the obtained materials, such as ILs could offer a carbon source to prepare fluorescence CDs [128] and in Chen’s study, the IL provided a fluorine-rich environment and helped to expose (001) facets of TiO_2_ [60].

## 7. Conclusions

The combination of microwave irradiation and ionic liquids (ILs) paves a powerful route towards high efficiency and low toxicity processes. In this review, applications of ILs in conjunction with microwave irradiation to prepare different inorganic, polymeric, carbon- and biomass-based materials were highlighted. ILs can be used as microwave absorbers, solvents, additives, templates and reactants. As ionic conductors, ILs absorb microwaves, which benefits heating processes. As solvents, ILs are low toxicity and, at the same time, are outstanding for cellulose dissolution. When used as additives, ILs can broaden molecular weight distribution of the resulting polymers. As templates, different types of ILs with different doses could control morphologies and properties of the resulting nanostructures. As reactants, ILs act as abundant carbon and nitrogen sources. Furthermore, some researchers suggested that ILs are recyclable and can be reused without loss of their performance.

Despite numerous achievements, several limitations still exist in the development of MAIL technology. First, only limited numbers of ILs were applied to prepare common materials, such as [Bmim]Cl and [Bmim][BF_4_]. Development and application of other none toxic types of ILs still need further scientific exploration. Second, there is still a lack of knowledge on the exact mechanism between microwaves, ILs and nanostructures or polymers. Thus, future research should focus on providing more experimental examples for better understanding these interactions. Third, recyclability of ILs has very important economic as well as environmental benefits for current and future technologies. Yet, it is still challenging to comprehend all the details of the processes and reactions related to ionic liquid recovery process. Finally, it is also essential to develop the large-scale industrial application of this method in the future.

In summary, this review provided literature-supported evidence of advantages of synthesis techniques involving microwave irradiation of IL-containing systems to obtain various functional materials. The goal of this review paper was not only to convey remarkable recent developments of this synthetic technology, but also to inspire the readers to study this novel and effective route for the new materials preparation even further.

## Figures and Tables

**Figure 1 nanomaterials-09-00647-f001:**
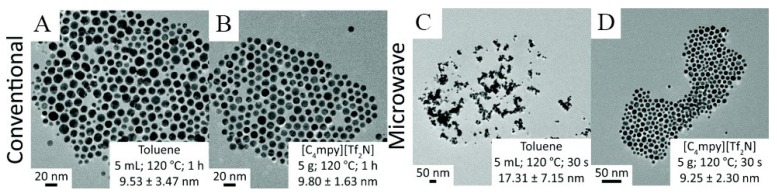
Scheme (**A**–**D**) representative transmission electron microscopy (TEM) images of Au nanoparticles (NPs) synthesized using conventional and microwave methods carried out in either toluene (panels A and C) or [C_4_mpy][Tf_2_N] (panels B and D) [43].Copyright Royal Society of Chemistry, 2018.

**Figure 2 nanomaterials-09-00647-f002:**
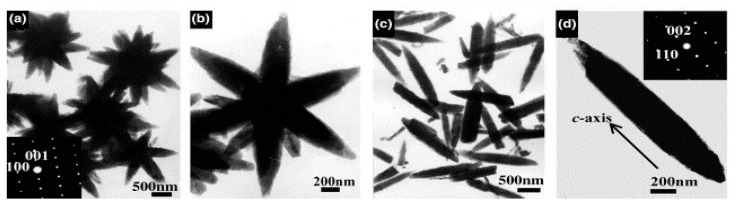
TEM micrographs: (**a**,**b**) Flowerlike ZnO; (**c**,**d**) needlelike ZnO [51]. Copyright Elsevier, 2004.

**Figure 3 nanomaterials-09-00647-f003:**
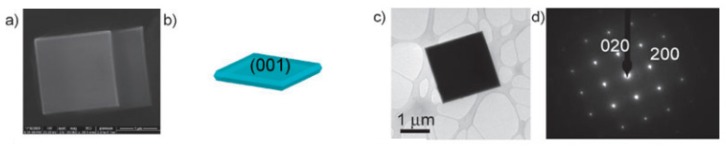
Field emission scanning electron microscope (FESEM) image (**a**), simulated model (**b**), TEM image of a representative anatase single crystal recorded along the (001) axis (**c**), selected area electron diffraction (SAED) pattern and (**d**) high resolution transmission electron microscope (HRTEM) image [59]. Copyright Royal Society of Chemistry, 2009.

**Figure 4 nanomaterials-09-00647-f004:**
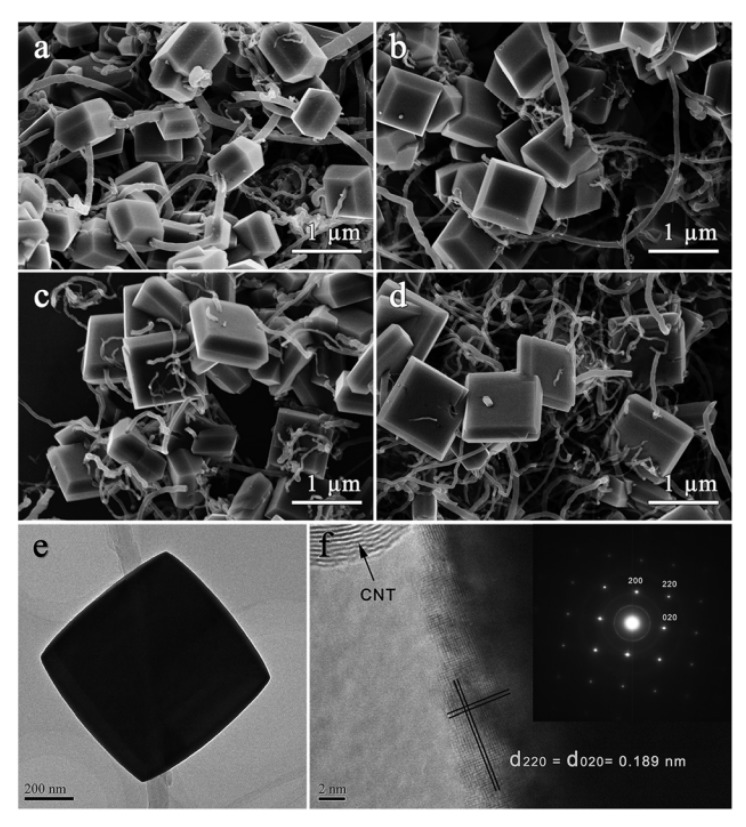
FESEM images of carbon nanotubes (CNTs)–TiO_2_ (different concentrations of TiCl_3_) (**a**–**d**), TEM (**e**) and HRTEM (**f**) image of CNTs–TiO_2_-4 (0.5M TiCl_3_), together with the CNTs–TiO_2_-4 SAED image (inset) [62]. Copyright Royal Society of Chemistry, 2016.

**Figure 5 nanomaterials-09-00647-f005:**
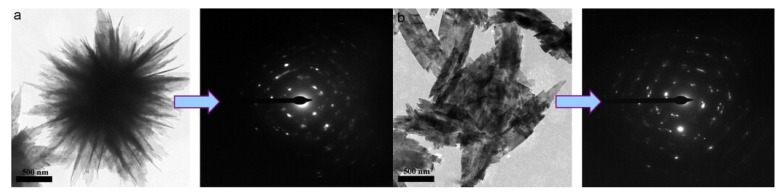
TEM and SAED images of flower-like and leaf-like CuO nanosheets prepared in a reflux system at 80 °C for 10 min under microwave irradiation, (**a**) 2 mL ionic liquid 1-octyl-3-methylimidazolium trifluoroacetate ([Omim]TA) and (**b**) 3 mL ionic liquid [Omim]TA. Scale bar: (**a**) 500 nm and (**b**) 500 nm [65]. Copyright Elsevier, 2009.

**Figure 6 nanomaterials-09-00647-f006:**
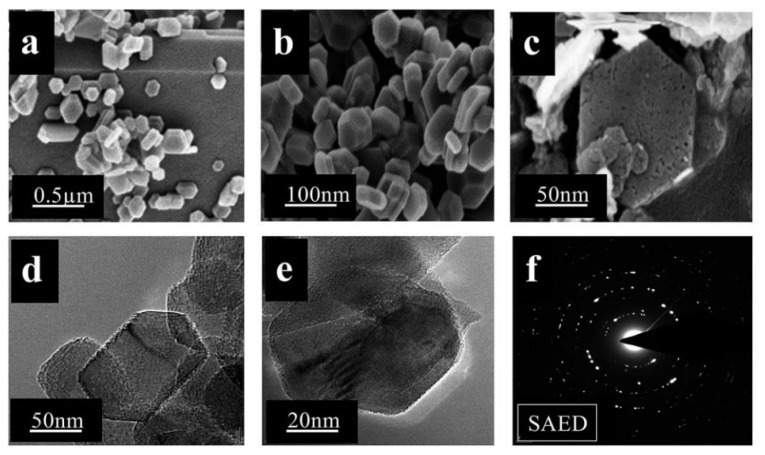
FESEM (**a**–**c**) and FETEM (**d**,**e**) images, and SAED pattern (**f**) of MgO nanohexagons obtained in bis(3-methyl imidazolium-yl) butane dichloride [74]. Copyright Royal Society of Chemistry, 2006.

**Figure 7 nanomaterials-09-00647-f007:**
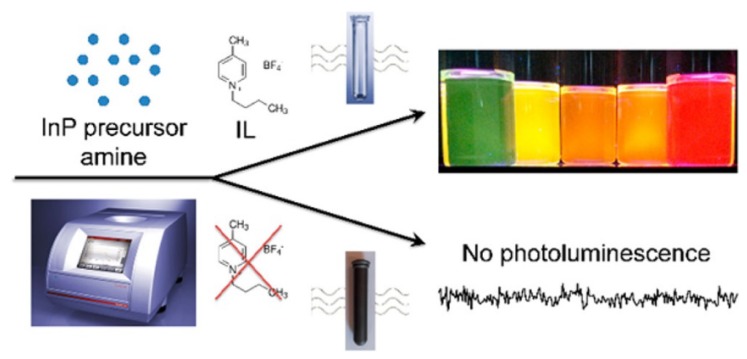
Scheme of microwave-assisted synthesis of InP nanocrystals in ionic liquid (IL) with color-tunable luminescence [77].Copyright American Chemical Society, 2017.

**Figure 8 nanomaterials-09-00647-f008:**
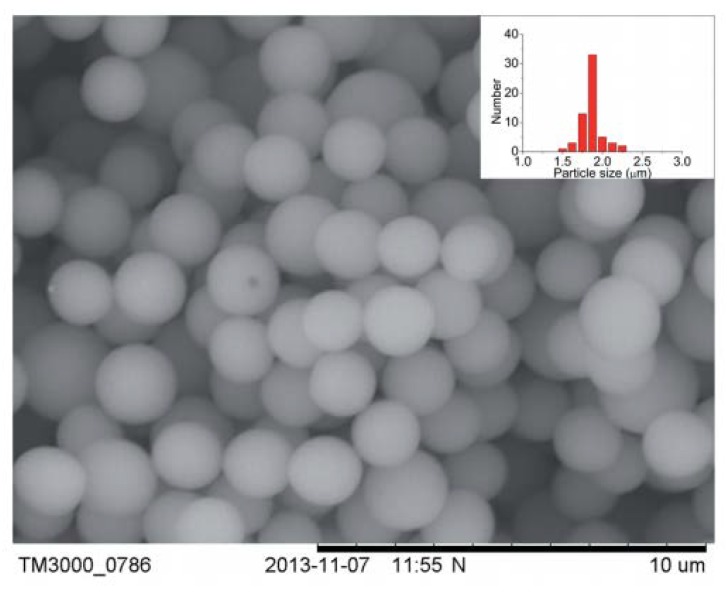
SEM image and size distribution (inset) of P[2-(methacryloyloxy)ethyl]trimethylammonium^+^ bis(tri-uoromethane sulfonyl)imide^−^) (P[MTMA][TFSI]) particles [112]. Copyright Royal Society of Chemistry, 2014.

**Figure 9 nanomaterials-09-00647-f009:**
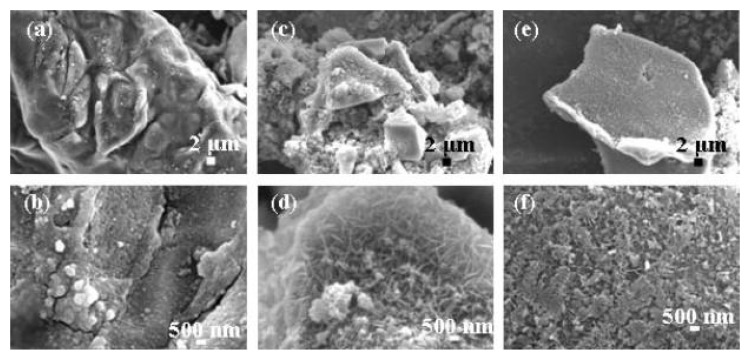
SEM micrographs of the cellulose/calcium silicate nanocomposites synthesized in (**a**,**b**) the starting ionic liquids, (**c**,**d**) the once-recycled ionic liquids, and (**e**,**f**) the twice recycled ionic liquids [137]. Copyright Elsevier, 2011.

**Figure 10 nanomaterials-09-00647-f010:**
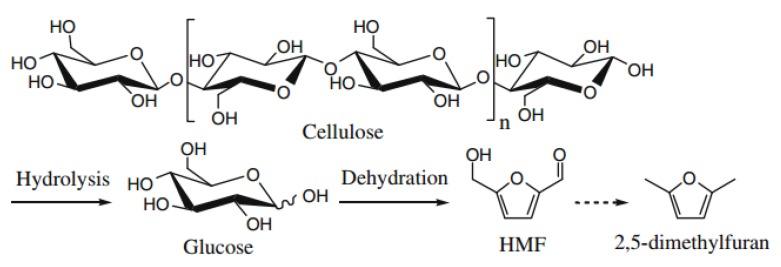
Schematic illustration of the steps for cellulose to HMF [148]. Copyright Elsevier, 2009.

**Table 1 nanomaterials-09-00647-t001:** An updated review of other metal structures prepared by microwave-assisted ionic-liquid (MAIL) methods reported in the literature.

Nanomaterial Properties.	ILs	Microwave Condition	Applications	Ref.
One-dimensional (1D) and 2D mesoporous nickel cobaltite (NiCo_2_O_4_) rods and sheets	[Bmim][BF_4_]	100 °C for 10 min	Efficient electrode material for supercapacitors	[80]
Metal-fluoride NPs, (MFx-NPs) with M = Fe, Co, Pr, Eu	[Bmim][BF_4_]	Irradiated for 10 min (Co) or 15 min (Fe, Pr, Eu) at a power of 50 W to a temperature of 220 °C	Cathode material for lithium-ion batteries	[81]
Nanoparticle morphology of Sb_2_Te_3_	1-alkyl-3-methylimidazolium and 1,3-dialkylimidazolium-based ILs	30 s at 100 °C, subsequently for 5 s at 150 °C and finally for 5 min at 170 °C	Thermoelectrics	[82]
Hierarchical microcube-like BiOBr	1-hexadecyl-3-methylimidazolium-bromide	160 °C	/	[83]
Sr_1−x_Ba_x_SnO_3_ Perovskite	[C_4_mim][Tf_2_N]	10 min at 85 °C	Photocatalytic applications for the hydroxylation of terephthalic acid	[84]
Co_2_P/CNTs	Tetrabutylphosphonium chloride ([P_4,4,4,4_]Cl), trihexyl(tetradecyl)phosphonium chloride ([P_6,6,6,14_]Cl)	Microwave oven for 6 min	Hydrogen evolution	[85]
SnSe_x_ NDs@rGO	[Bmmim]Cl	120 °C for 5 min followed by 180 °C for 55 min	Electrochemical application	[86]
MoS_2_/BiOBr	[C_16_mim]Br	140 °C for 10 min	Photocatalyst	[87]

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
