# Peer review of "New Developments in Material Preparation Using a Combination of Ionic Liquids and Microwave Irradiation"

_nanomaterials, 2019, doi:10.3390/nano9040647_

Round 1
Reviewer 1 Report
This review aims to provid literatures-supported evidence of advantages of synthesis techniques involving microwave irradiation of IL-containing systems to obtain various functional materials. This reviewer feels that it may be a useful paper but the Green chemistry increasing emphasis for IL is misplaced as none of the Fluorine containing IOL will be considered friendly. They should talk about amino acid-based IL as some are now known as described by Robin Rogers. Combination of microwave irradiation and ionic liquids (ILs) paves a powerful route towards this trend, is the good comment and this reviewer concurs.
The statement needs to be modified to eliminate the word. Green’ here in the sentence in conclusion and elsewhere in the text: “This method is not only regarded as “green” and economic alternative. It also decreases reaction time and increases the reaction efficacy”.
Some relevant references should be added in the introduction section:
where alternate energy systems such as ultrasound has been used for IL preparation: Solvent-free Sonochemical Preparation of Ionic Liquids. Org. Lett., 4, 3161-3163 (2002
as well as aqueous systems generating unique structures from salts: Synthesis and Applications of Micro-Pine Structured Nano-Catalyst: Chem. Commun., 6318, (2008) and Self-Assembly of Metal Oxides into Three-Dimensional Nanostructures: Synthesis and Application in Catalysis: ACS Nano, 3, 728-736 (2009)
and where unique systems have been used for synthesis and nanocatalysts, together: Microwave-assisted Reductive Amination with Aqueous Ammonia: Sustainable Pathway Using Recyclable Magnetic Nickel-Based Nano-catalyst: ACS Sustain. Chem. & Eng., 7, (2019) DOI: 10.1021/acssuschemeng.8b06054
General overview-type article: Alternate Energy Input: Mechanochemical, Microwave and Ultrasound-assisted Organic Synthesis: Chem. Soc. Rev., 41, 1559-1584 (2012).
Author Response
Dear Reviewer:
On behalf of my co-authors, we thank you very much for giving us an opportunity to revise our manuscript, we appreciate very much for the positive and constructive comments and suggestions on our manuscript.
Point 1: This reviewer feels that it may be a useful paper but the Green chemistry increasing emphasis for IL is misplaced as none of the Fluorine containing IOL will be considered friendly.
Response 1: Regarding to this reviewer's suggestion, we considered this question carefully and we reconsider the word “green” and “friendly” in our manuscripts (Line 12, 24, 40, etc.). Although part of ionic liquids contain fluorine, but they really played an important role in the nanostructure preparation. For instance, [BF4]- in [Bmim][BF4] is unstable in aqueous solution at higher temperature, and will decompose thermally and hydrolyse slowly under the appropriate conditions to release F- ions, which definitely be a drawback for synthesis but is extremely helpful for the synthesis of fluoride nanomaterials with promising performance. Compared with the common fluoride such as HF or NaF, [Bmim][BF4] acting as a fluoride source is more environmentally friendly and operationally safe. At the same time, microwave is an efficient and safe heating method. So, we believe that the MAIL technology still has great potential for the development of materials syntheses for cleaner process.
Beside this, we added some more information the development of ionic liquids (line 44-57).
Point 2: Some relevant references should be added in the introduction section.
Response 2: As the reviewer suggested that we read the literatures and added more information in the introduction section (line 58, 76, 82).
We appreciate your warm work earnestly and hope that the correction will meet with approval. Once again, thank you very much for your comments and suggestions.
Reviewer 2 Report
The present manuscript reviews materials syntheses by using ionic-liquids in combination with microwave irradiation. The materials described here are inorganic nanoparticles, polymers, carbon-derived materials, and cellulose-based composites. In these synthetic methods, utilization of ionic liquids is crucial as microwave absorbers, solvents, additives, templates, and reactants. Various examples are represented in the manuscript, which may be helpful for many researchers. Although the detailed mechanism of the microwave-assisted ionic-liquid methods is still unclear, the methods have great potential for the development of much greener process of materials syntheses. The reviewer thinks that this review paper is acceptable for publication after minor revisions.
1. The “Figure 2” in line 109 will be missing. Please check it.
2. As a comment, typical mechanism of the microwave-assisted ionic-liquid methods could be reviewed more in detail. Schematic diagrams of the mechanism may be helpful for the readers to understand the chemistry of the microwave-assisted ionic-liquid methods.
3. There are some mistypes in the body text, which should be checked.
Author Response
Dear Reviewer:
On behalf of my co-authors, we thank you very much for giving us an opportunity to revise our manuscript entitled “New Developments in Material Preparation Using Combination of Ionic Liquids and Microwave Irradiation”. We appreciate very much for the positive and constructive comments and suggestions on our manuscript.
Point 1: The “Figure 2” in line 109 will be missing. Please check it.
Response 1: We are very sorry for our negligence of this detail. The missing Figure didn’t provide more information compared with Figure 1, so we decide to delete it. But we forgot delete it in the main text.
Point 2: As a comment, typical mechanism of the microwave-assisted ionic-liquid methods could be reviewed more in detail. Schematic diagrams of the mechanism may be helpful for the readers to understand the chemistry of the microwave-assisted ionic-liquid methods.
Response 2: As the reviewer suggested that we summarized several mechanisms involved in microwave-assisted ionic-liquid methods for materials preparation (line 616-650). As there are different mechanisms in different processes, so we are sorry we can't provide a schematic diagrams exactly.
Point 3: There are some mistypes in the body text, which should be checked.
Response 3: We are very sorry for our negligence of this detail, we’ve revised these mistypes in our revised manuscript (line 159, 210 etc.).
We appreciate your warm work earnestly and hope that the correction will meet with approval. Once again, thank you very much for your comments and suggestions.
Reviewer 3 Report
The authors summarize recent developments on the use of microwaves and ILs for the synthesis of inorganic nanomaterials, polymers, carbon-derived materials and cellulose-based composites.
The manuscript is written in good English but typing errors have to be corrected as well as the references (years not in bold sometimes, initial and final pages, …)
The introduction devoted to the use of microwaves and/or ILs is quite complete and relatively well described with quite old references except one concerning the analysis of proteins.
Successively, the preparation of inorganic nanomaterials (metal, metal oxides or complexes), of polymers, carbon-based and cellulose-based materials was described. The parts concerning the inorganic compounds and the carbon-based derivatives are well described and complete with appropriate chosen examples. For the polymers, it could be interesting to add some works concerning the compatibilization of polymers or copolymers through ILs and appropriate related references. Same remark could be also given for the part related to cellulose-based materials. Here the works concerning the transformation (dissolution or defragmentation) of the raw materials have to be added; the example of the 5-Hydroxymethylfurfural is given but it’s not the only one. This part has to be seriously revised and performed.
In conclusion, this manuscript could not be accepted for a publication in Nanomaterials.
Author Response
Dear Reviewer:
On behalf of my co-authors, we appreciate very much for the constructive comments and suggestions on our manuscript entitled “New Developments in Material Preparation Using Combination of Ionic Liquids and Microwave Irradiation”. We revised our manuscript carefully and we sincerely hope you could reconsider your comment about this manuscript.
Point 1:The manuscript is written in good English but typing errors have to be corrected as well as the references (years not in bold sometimes, initial and final pages, …)
Response 1: We are very sorry for our negligence of those mistakes, we’ve revised these mistypes in our revised manuscript and references (line 159, 210, 705 etc.).
Point 2: The introduction devoted to the use of microwaves and/or ILs is quite complete and relatively well described with quite old references except one concerning the analysis of proteins.
Response 2:We added some recent literatures to perfect the introduction (Ref. 9, 13, 30, 33). We are very sorry for we do not understand the meaning of “the analysis of proteins” as this manuscript does not involve in proteins science.
Point 3: For the polymers, it could be interesting to add some works concerning the compatibilization of polymers or copolymers through ILs and appropriate related references. Same remark could be also given for the part related to cellulose-based materials. Here the works concerning the transformation (dissolution or defragmentation) of the raw materials have to be added; the example of the 5-Hydroxymethylfurfural is given but it’s not the only one. This part has to be seriously revised and performed.
Response 3: Thanks for your constructive suggestion. We added some relevant works about biomass-based materials, furfural and sugar (Line 582-615). We know that ionic liquid and microwave have numerous applications in biomass hydrolysis. As this review focused on various materials preparation with the combination of ionic liquid and microwave, we think regard cellulose-Basesd composites, 5-Hydroxymethylfurfural, furfural and sugar as typical examples is enough for this part.
For the polymers, we are sorry we didn’t found enough references about compatibilization of polymers and we had already summarized the application of this method in copolymers (line 366-388). Furthermore, typical mechanism was reviewed more in detail which could provide more information for the readers (Line 617-650) .
Those microwave-assisted ionic-liquid methods have great potential for the development of materials syntheses, but there are few reviews about this method. In this text, various examples are represented, which may be quite helpful for many researchers. Based on the above revisions, we sincerely hope you could reconsider your comment about this manuscript.
We appreciate your warm work earnestly and hope that the correction will meet with approval. Once again, thank you very much for your comments and suggestions.
Reviewer 4 Report
The manuscript is written clearly and corresponds to the journal with its contents. Authors should correct claims from the introduction that ionic liquids have low toxicity and "very environmentally friendly" in accordance with recent scientific results.
Author Response
Dear Reviewer:
On behalf of my co-authors, we thank you very much for giving us an opportunity to revise our manuscript entitled “New Developments in Material Preparation Using Combination of Ionic Liquids and Microwave Irradiation”. We appreciate very much for the positive and constructive comments and suggestions on our manuscript.
Point 1: Authors should correct claims from the introduction that ionic liquids have low toxicity and "very environmentally friendly" in accordance with recent scientific results.
Response 1: Regarding to this reviewer's suggestion, we considered this question carefully and we reconsider the word “green” and “friendly” in our manuscripts (Line 12, 24, 40, etc.). Although part of ionic liquids contain fluorine, but they really played an important role in the nanostructure preparation. For instance, [BF4]- in [Bmim][BF4] is unstable in aqueous solution at higher temperature, and will decompose thermally and hydrolyse slowly under the appropriate conditions to release F- ions, which definitely be a drawback for synthesis but is extremely helpful for the synthesis of fluoride nanomaterials with promising performance. Compared with the common fluoride such as HF or NaF, [Bmim][BF4] acting as a fluoride source is more environmentally friendly and operationally safe. At the same time, microwave is an efficient and safe heating method. So, we believe that the MAIL technology still has great potential for the development of materials syntheses for cleaner process.
Beside this, we added some more information the development of ionic liquids (line 44-57).
We appreciate your warm work earnestly and hope that the correction will meet with approval. Once again, thank you very much for your comments and suggestions.
Round 2
Reviewer 1 Report
The paper has definitely improved fair bit. However, some English words are misspelled, especially newly added words. As example line 189, it should be: combined
line 209, should have: structure; line 244, should correct: Properties; line 411: microwave-assistance; line 668: should have correctly written: Non-toxic, among others.
So, a quick check of spelling will do the job.
Author Response
Dear Reviewer:
On behalf of my co-authors, we appreciate your warm work earnestly and thank you very much for giving us an opportunity to revise our manuscript. We are very sorry for our negligence of those detail, we’ve carefully revised these mistypes in our revised manuscript. We hope that the correction will meet with approval.